# Reward System Dysfunction and the Motoric-Cognitive Risk Syndrome in Older Persons

**DOI:** 10.3390/biomedicines10040808

**Published:** 2022-03-30

**Authors:** Fulvio Lauretani, Crescenzo Testa, Marco Salvi, Irene Zucchini, Beatrice Lorenzi, Sara Tagliaferri, Chiara Cattabiani, Marcello Maggio

**Affiliations:** 1Department of Medicine and Surgery, University of Parma, 43126 Parma, Italy; crescenzo.testa@unipr.it (C.T.); marco.salvi@unipr.it (M.S.); irene.zucchini@unipr.it (I.Z.); beatrice.lorenzi@unipr.it (B.L.); sara.tagliaferri@unipr.it (S.T.); marcellogiuseppe.maggio@unipr.it (M.M.); 2Cognitive and Motor Center, Medicine and Geriatric-Rehabilitation Department of Parma, University-Hospital of Parma, 43126 Parma, Italy; chiarac2004@libero.it; 3Geriatric Clinic Unit, Geriatric-Rehabilitation Department, University Hospital, 43126 Parma, Italy

**Keywords:** reward system, dopamine, glutamate, Alzheimer’s disease, Parkinson’s disease, older persons, cognitive and motoric risk syndrome

## Abstract

During aging, many physiological systems spontaneously change independent of the presence of chronic diseases. The reward system is not an exception and its dysfunction generally includes a reduction in dopamine and glutamate activities and the loss of neurons of the ventral tegmental area (VTA). These impairments are even more pronounced in older persons who have neurodegenerative diseases and/or are affected by cognitive and motoric frailty. All these changes may result in the occurrence of cognitive and motoric frailty and accelerated progression of neurodegenerative diseases, such as Alzheimer’s and Parkinson’s diseases. In particular, the loss of neurons in VTA may determine an acceleration of depressive symptoms and cognitive and motor frailty trajectory, producing an increased risk of disability and mortality. Thus, we hypothesize the existence of a loop between reward system dysfunction, depression, and neurodegenerative diseases in older persons. Longitudinal studies are needed to evaluate the determinant role of the reward system in the onset of motoric-cognitive risk syndrome.

## 1. Introduction

For years, the clinician’s approach to the elderly patient has been steeped in a dichotomy: motor dysfunction or cognitive impairment.

Thus, in the geriatric assessment units, outpatients were evaluated separately for cognitive impairments and motor disturbances.

Recently this “traditional” custom has been questioned by Joe Verghese and colleagues who have identified in the Motoric Cognitive Risk syndrome (MCR) a clinical tool to be provided to clinicians.

This syndrome allows an integrated approach to the assessment of patients at risk of dementia and to overcome the paradigm of the dichotomy between motor and cognitive disorders [1].

As a consequence, this translates into a better stratification of patients at risk of dementia, useful for modifying the progression curve towards disability.

This innovation is absolutely necessary given the recent forecasts on the increase in the incidence and prevalence of people living with dementia that foresees a dramatic picture for 2050 [2].

People living with dementia are expected to increase from 57.4 (95% CI 50.4–65.1) million in 2019 to 152.8 (95% CI 130.8–175.9) million in 2050 [2].

This enormous increase in the number of cases of patients with dementia requires a change of approach to the care of these patients, especially in the stages preceding overt dementia.

The need for an integrated approach to prevent the conversion of patients with pre-dementia by going to work on modifiable risk factors seems intuitive.

The appearance of Motoric Cognitive Risk Syndrome, a clinical syndrome that combines slow walking speed and subjective cognitive disorders in the clinical arena, is very welcome.

It was introduced in 2013 by Verghese and colleagues [3] and since then increasing evidence has accumulated on its epidemiology, pathogenesis, and the increased risk in patients who meet the diagnostic criteria of this syndrome of converting to overt dementia [4].

In summary, despite the different diagnostic criteria used in the various studies, the average prevalence of Motoric Cognitive Risk Syndrome in the world population is around 10% [5] with a risk of converting into dementia that is increased by about three times compared to those who do not meet the diagnostic criteria [6].

Even if the level of evidence is not yet sufficiently high, it has now been established that lack of sociability and depression are very important risk factors which, if intercepted, could modify the epidemiological curve of the incidence of new cases of dementia in the world [7]. Depression and systems to maintain well-being could be influenced by the reward system and a link between them may synergistically produce a deflection of cognition in older persons.

In this study we will investigate the rewards system in humans and elderly patients, highlighting the links between the rewards system, the motoric cognitive risk syndrome and depression in the elderly patient where, however, it is still possible to intervene to at least slow the progression towards cognitive decline. To do this we hypothesized a loop between the reward system, depression, and cognitive decline in elderly patients.

## 2. Reward System in Humans and in Older Persons

The reward system is probably the very reason why evolution has endowed organisms with brains. Species with brains are able to obtain better rewards in the environment necessary to survive [8]. The brain needs to recognize a reward, and a reward is such not because of its physical properties, but because of the inducible behaviors [8].

We can categorize the rewards into primary and non-primary. The primary rewards ensure the propagation of the most suitable genes to the environment that selected them, the survival of individuals and their reproduction. These types of rewards are, for example, food and liquids that contain the nutrients necessary contribute to the balance and the propagation of the species.

The non-primary rewards are strictly linked to the primary rewards, and, like the former, they contribute to a better adaptation of the individual to the environment. However, unlike the primary rewards, they are specific for everyone.

Due to the modality and temporality in which its action potential is expressed, the dopaminergic neuron is the first interlocutor when analyzing the reward system in the human being [9]. Behavioral studies have shown that dopaminergic projections to the striatum and frontal cortex are a critical hub for the flow of information about rewards [9].

In vivo studies have investigated selective lesions in different components of the dopaminergic system, and the systemic or intracerebral administration of dopaminergic agonists or antagonists and studies on substances of abuse [10].

Dopaminergic neurons have distinct firing rates activated by a wide variety of reward stimuli regardless of the type of reward. Dopaminergic neurons, a minority compared to other neuronal groups within the brain, are a very heterogeneous group of cells, both anatomically and functionally.

Most of the cell bodies of dopaminergic neurons are located in the diencephalon, midbrain and olfactory bulb. It is very important to underline the role of a group of neurons located at the level of the substantia nigra and the ventral tegmental area. This group of neurons is important for its projections at the level of the striatum, and particularly at the caudate nucleus, at the putamen and at the level of the ventral striatum. Additional projections are directed to the nucleus accumbens, one of the brain hotspots of the sensation of pleasure, directly linked to the reward system [11].

The main dopaminergic pathways within the brain consist of projections from dopaminergic cell bodies located at the level of the substantia nigra, the ventral tegmental area and the arcuate nucleus of the hypothalamus; dopaminergic projections depart from these hubs which give life to the nigrostriatal, mesolimbic, mesocortical and tuberoinfundibular pathways [12] (Figure 1).

Each of these pathways, with its own specificity, is integrated into a macrosystem governing and connecting motor and cognitive functions. For example, the projections of the nigro-striatal pathway deeply influence the motor and motivational aspects of behavior [13]. As far as the mesolimbic pathway is concerned, its dysfunctions are directly linked to the pathogenesis of addiction to psychostimulant substances. However, there is recent evidence that the integrity of mesolimbic pathway is also involved in the reward system when, for example, someone eats some pleasant food [14].

The meso-cortical pathway, which is also decisively involved in the development of emotional and behavioral processes related to the reward system, offers further reflection. This pathway is not purely dopaminergic but is also connected with the glutamate neurotransmitter system [15]. This evidence lays the biological foundations of a reward system in which, even if the role of dopaminergic neurons is certainly fundamental, the interconnection with other neurotransmitter systems is nevertheless important for the correct functioning of the reward system itself.

The tubero-infundibular pathway originates from the arcuate nucleus of the hypothalamus and connects it to the median eminence of the neurohypophysis. This dopaminergic pathway, albeit less studied, is integrated into the reward system and lays the biological foundations that connect the reward system to the maintenance of the homeostatic functions of the individual [16]. The neurohypophysis is in fact the fundamental hub through which the brain regulates the homeostasis of peripheral organs and is directly influenced by the scarcity (hunger) or abundance (satiety) of metabolites in the bloodstream [17].

The biological substrate of such a heterogeneity of connections and functions, always perfectly integrated in the macrosystem of rewards, offers an equal heterogeneity in terms of receptors.

Five main types of dopaminergic receptors are described. D1 and D2 dopamine receptors are the most expressed receptor types in the central nervous system. Their abandonment places them as a classifying example. The class of dopamine D1 receptors (D1 and D5 receptors) is coupled to a stimulatory G protein which increases the intracellular concentration of cyclic AMP. The class of dopamine D2 receptors (D2, D3 and D4) is instead coupled to an inhibitory G protein that inhibits the production of cAMP. The picture is complicated because each receptor type corresponds to a subclass that interacts with its own intracellular signaling [18]. If we more specifically analyze each receptor subtype, it is clear that the connections between the dopaminergic system and the other neurotransmitter systems are evident.

For example, D1 transmission is involved in the regulation of GABAergic, glutamatergic and cholinergic neurotransmission. D2 receptors function as self-receptors and have been found in both the dendritic and axonal compartments where regulators of dopaminergic transmission itself. D3 receptors are practically ubiquitous in dopaminergic neurons and contribute, also as self-receptors, to the regulation of dopamine release [19].

D4 receptors are abundantly expressed in the frontal lobes, hence their involvement in higher cognitive functions. Finally, the D5 receptors are mainly expressed at the hypothalamic level, the hub of homeostatic control in the organism [20]. But the dopaminergic system is not only based on receptor endings; dopaminergic transmission is profoundly influenced by proteins that modulate the presence of dopamine at the level of the intrasynaptic space after it has been released.

The two main actors of this modulation include catecholoxymethyltransferase (COMT), which catabolizes the released dopamine at the level of the synaptic space and the dopamine transporter (DAT) whose function is fundamental. DAT modulates the intensity of the signal at the level of the synaptic space, and dopaminergic transmission itself.

Like any protein, COMTs and DATs are also encoded by genes that respond to allelic variations of all other genes. It has been demonstrated, through in vivo functional magnetic resonance studies, that certain alleles correspond to activations of different intensities at the level of the components of the reward system.

This suggests that the anticipation and reception of rewards (there are a multitude of non-primary rewards, all individual specific) in different people is also determined by the genotypic set-up [21].

The physiological basis of the functioning of biological systems can often be deduced from the study of diseases. Models of Parkinson’s disease investigating experimental lesions of dopaminergic neurons and side-effects of typical antipsychotic drugs, suggest a series of very interesting considerations on the functioning of the dopaminergic system.

Deficits in dopaminergic transmissions are linked both to motor (akinesia, rigidity, and tremor at rest) and cognitive deficits (loss of attention and learning deficit) and to apathetic states (depression, reduced response to emotional stimuli). The administration of dopamine precursors, capable to improve symptoms, does not restore the phasic transmission patterns of dopaminergic neurons [9].

The dopaminergic transmission makes use of two components, a phasic one which processes appetitive and alert information and a tonic, broader one, connected to the different dopaminergic systems previously analyzed.

Tonic dopaminergic transmission is based on low-dopamine doses at the level of the different dopaminergic areas. Low dopamine levels are generally detected by auto-receptors, especially D2, and modulate the functioning of the brain area where this transmission occurs. Dopamine levels are finely balanced by several modulators, such as COMT and DAT [22,23].

Each area involved in the reward system has its own specific tonic dopaminergic activity [24]. This “dopaminergic micro-environment” is finely modulated and the dopaminergic brain areas probably need a specific level of dopamine to best perform their function [9]. Upward (schizophrenia) or downward (Parkinson’s disease) excesses of tonic dopamine levels underlie a dysfunction of the involved dopaminergic area.

Ultimately, there is a fundamental role of the reward system, based on the functionality and mode of discharge of dopaminergic neurons, as a component of learning and high cognitive functions.

Probably the most important reason why such a sophisticated system for predicting and receiving rewards has evolved is because of its facilitating action on ability learning and the relationship with the environment. This makes the reward system a mainstay in the individual’s interaction with his surroundings and peers [8,25].

The dopaminergic system is particularly vulnerable to advancing age. This is attributable to chemical reactions that lead to the synthesis of dopamine. This process includes an unstable oxido-reductive balance which is sensitive to the accumulation of free radicals [26,27].

Given the importance of the binomial tonic and phasic functions of the dopaminergic system, both preclinical and clinical studies have shown a deterioration of the components of the dopaminergic system with advancing age [28,29].

The decrease in the expression of dopaminergic receptors and the opposite change in dopamine synthesis is individual-specific with huge individual vulnerability [28,30]. It is estimated that in humans there is a loss of about 10% of dopaminergic neurons every decade, and this decline in dopaminergic function sets the stage for both motor and cognitive impairments [4,31,32,33,34].

Nuclear medicine studies have shown that the expression of D1 and D2 receptors, as well as the expression of DAT are reduced when analyzing the nigrostriatal pathway during aging [35,36,37]. It should be noted that these alterations in the microscopic components of the dopaminergic system, in certain cases, do not lead to a clear downregulation of the dopaminergic system.

When analyzing the reward system in the elderly, indeed in some cases an increase in dopaminergic transmission is even observed [38]. This evidence supports the hypothesis that, more than an increase or a reduction of the dopaminergic signal, the balance in the synaptic space is crucial for the correct functioning of the brain areas. This hypothesis puts the dopaminergic signal in contact with the rest of the brain neurotransmitter systems.

After analyzing the nigrostriatal system, one cannot but take into consideration neuromelanin, a dark colored pigment that typically accumulates in the cells of the substantia nigra, to which it gives its name. Levels of neuromelanin increase with age and accumulate in the lysosomal organelles of dopaminergic cells. When neuromelanin levels build up to the point that they occupy 50% of the cytoplasmic space in dopaminergic cells, the tendency of α-synuclein to form cell-damaging protein aggregates is increased [39,40]. The accumulation of alfa-sinuclein is one of the key pathogenetic components of Parkinson’s disease [41]. In addition to the nigro-striatal pathway, the mesolimbic and mesocortical pathways are also affected by age-related dysfunctions [42].

In the reward system, dopaminergic neurons at the level of the ventral tegmental area project to nucleus accumbens and prefrontal cortex. These circuits, as already seen, lay the foundations for the identification and reception of rewards [43,44].

In particular, the dopaminergic stimulation of medium spiny neurons at the level of the nucleus accumbens is fundamental for selecting motivational stimuli connected with rewards [45]. Age-related dysfunctions of this complex system have already been demonstrated with neuroimaging methods [38].

With advancing age and during the development of neurodegenerative diseases such as Parkinson’s and Alzheimer’s, there is a marked dysfunction of the dopaminergic system and the incorrect processing of reward stimuli [46,47].

## 3. The Contribution of Reward System to the Motoric Cognitive

### 3.1. Risk Syndrome in Older Persons

The brain networks that integrate the functions of different cortical domains are the cornerstone of the correct functioning of the central nervous system. These networks are based on the presence of plastic structural connections, and the more these connections are developed, the healthier the central nervous system is.

These brain networks, in healthy individuals, form circuits that involve the motor system and the cognitive system. The same development in the cortex, seat of the highest cognitive functions, of areas that control movement lays the foundations for an integrated vision of the motoric-cognitive system [48,49].

The rewards system enters into this context and its integrity is essential for the proper functioning of the whole system. One of the key components of this system is mirror neurons [50].

Several studies have shown that the presence or absence of rewards can modify the excitability of the motor cortex both when a movement is made and when a movement is observed, with different outcomes on the learning of motor tasks [51,52]. There is increasing evidence that the learning process of motor tasks is improved in the presence of rewards [53].

The best described reward network is the mesolimbic pathway, which starts from the ventral tegmental area and affects the nucleus accumbens of the ventral striatum, some nuclei of the terminal stria, the amygdala and the hippocampus [54].

Accumulating evidence highlights an exchange of information between the pathways of the reward system and the functional pathways of mirror neurons [55]. Even in experiments on primates, some neurons of the mirror neuron system distinguish a reward from a non-reward regardless of the receiver [55]. When these connections fail, together with social isolation and other age-related risk factors, an anhedonic state occurs. This event can be interpreted as the proxy of reward system impairment [54].

The subsequent worsening of cognitive and motor functions leads to the Motoric Cognitive Risk Syndrome. Analyzing the brain networks of patients with Parkinson’s through multimodal neuroimaging methods, we can observe the hypoactivity of different cortical areas. One of these networks is the Inferior Parietal Cortex (IPC), which is hypoconnected with respect to the other cortical networks.

IPC represents an intersection of different networks and its dysfunction could represent the link between motor and cognitive deficits in patients with Parkinson’s where there is a primary dysfunction of the dopaminergic system [56].

Studies have shown that successful aging and maintenance of good cognitive abilities are based on efficient brain networks in elderly patients [57,58]. Although the pathogenesis of neurodegenerative diseases and dementia is still a source of widespread debate. there is increasing evidence indicating that in the elderly the maintenance or in any case the presence of more effective brain networks is associated with greater resilience to cognitive decline [59]. Functional neuroimaging studies with PET-amyloid in elderly patients corroborate the hypothesis of a diffuse deposition of amyloid determining dysfunction in several cognitive domains including executive functions, behavior and physical performance [60,61].

Studies have investigated the functional efficiency of brain networks, in the presence of a greater deposition of amyloid, with better efficient structural networks considered a proxy of increased resilience to cognitive decline [59]. Other evidence indicates that brain networks in common between motor and cognitive functions may be affected by neurodegeneration [62].

Patients with Motoric Cognitive Risk Syndrome show a smaller volume of total gray matter, cortical gray matter, premotor cortex, prefrontal cortex and dorsolateral segment of the prefrontal cortex compared to those who do not meet the diagnostic criteria for MCR [63]. 

Atrophy of the supplementary motor cortex, insular cortex and prefrontal cortex was found in a study of gray matter networks in patients with MCR [64]. Some of these data have been confirmed in a systematic review which shows that, in patients with MCR, there is a reduction of gray matter at the level of the premotor cortex in the prefrontal cortex [65].

Many of these brain areas are integrated into functional circuits whose networks are shared by the motor system (with the participation of the mirror neuron system), cognitive functions and the reward system.

The integration between these systems is still fundamental in the elderly patient, where the resilience to cognitive decline is based precisely on the integrity of the brain networks. In particular, in these individuals, dopaminergic function integrating many of these systems is fundamental in determining the amount of physical activity. In a very elegant study it was shown that elderly patients who express allelic variants of more effective dopamine receptors perform a greater amount of physical activity than those who express less effective receptor allelic variants [66].

All these data together [46,47] suggest that in older persons a dysfunction of the dopaminergic system predisposes the development of the Motoric Cognitive Risk Syndrome.

### 3.2. Reward System and Depressions in Older Persons

Being part of a community and being integrated into multiple social groups and activities is much more important than expected. The COVID 19 epidemic has taught us, once again, two concepts: the existence of a “social brain” and contact with others is a *sine qua non* condition for being healthy [67]. This “social activity”—the viewing and meeting of others—should be considered a reward [68].

It becomes more understandable why social isolation that often accompanies aging should be recognized as a very important risk factor for the onset of depression in the elderly.

We have already highlighted that a dysfunction of the dopaminergic system is an important risk factor for the development of anhedonia [54]. When there are functional impairments in the reward system the reward response (regardless of whether it is a primary reward or a non-primary reward) is ineffective. As a consequence, there is a reduction in the behavioral drive that leads to the pursuit of rewards, the development of depressive symptoms which are often accompanied by an apathetic phenotype [69].

Apathy is an affective state characterized by the loss of interest and indifference towards the surrounding world and it is a very frequent behavioral symptom in elderly patients. It should be emphasized that the severity of symptoms increases as cognitive decline progresses [70].

Apathy is already present in the young patients, and is less rare and more pronounced in the elderly patient [71].

The Iowa Gambling Task is a neuropsychological test designed to investigate reward decision making [72]. The Iowa Gambling Task has been recently used to examine the response to rewards of 60 non-demented older adults with major depression and 36 psychiatrically healthy older adults. Patients with apathy (quantified by the Apathy Evaluation Scale) had a more conservative behavioral response [73].

The authors of this study attributed this response to either a reduced motivation to seek a rewarding experience or a reduced sensitivity to rewards.

Neuroimaging studies have shown that the apathetic phenotype of major depression in the elderly is associated with a dysfunction of the cortical connections at nucleus accumbens, amygdala, caudate, putamen, globus pallidus and thalamus, the traditional areas of reward system [74].

A dysfunction of the reward system has been associated with the development of depression independent of apathy, age, and morbidity [54,75]. Depression is one of the most frequent non-motor symptoms in Parkinson’s disease and is even the onset symptom of the disease in some patients [76].

Moreover, patients with Parkinson’s disease display dysfunction in the reward system, particularly in the mesolimbic pathway of the dopaminergic system, which predisposes to depression [77,78,79,80,81]. A recent meta-analysis of 55 studies involving 2578 patients (1638 patients with Parkinson’s disease and 940 healthy controls) shows that Parkinson’s disease is characterized by a dysfunction in reward processing, with the involvement of dopaminergic circuits. This dysfunction may be the causal mechanism underlying neuropsychiatric disorders in Parkinson’s disease, such as depression [82].

The role of the dopaminergic system is certainly less important in the pathogenesis Alzheimer’s disease. However, animal models and human studies have shown that a dysfunction of the dopaminergic system may cause neuropsychiatric disorders including depression in patients with Alzheimer’s disease, [83,84]. Dysfunction of the dopaminergic system in patients with Alzheimer’s disease is also related to the apathetic phenotype [85].

A dysfunction of the reward system with depression seems to affect not only the course of neurodegenerative but also ischemic diseases. Post-stroke depression is a serious complication affecting approximately 30% of patients who have experienced an ischemic event [86]. Patients with post-stroke depression have a greater risk of long-term complications and a lower rehabilitation reserve than patients who do not develop this disorder [87].

Impairment in functional connections at the level of the reward system has been shown in ischemic patients. Through a connectome study derived from magnetic resonance, microstructural alterations were highlighted precisely at the level of the connections of the reward system with a pathophysiological pattern attributable to neuroinflammation [88]. The correlation between changes in the reward system and post-stroke depression suggests clinical implications in the clinical practice of stroke care pathway. In particular, this additional assessment could help the clinician to identify patients at higher risk of post stroke depression. A higher prevalence of patients with depression was also found in patients who meet the criteria for Motoric Cognitive Risk Syndrome [89,90,91].

When analyzing the pathophysiological mechanisms of depression, regardless of the setting, high levels of inflammatory cytokines are positively correlated with the incidence of depression [92,93].

Elevated plasma concentrations of inflammatory cytokines Interleukin 6 and C-reactive protein are also associated with a smaller volume of gray matter in non-demented patients [94] and reduced hippocampal volume [95]. Interestingly, a reduction in total gray matter and hippocampal atrophy is also associated with a higher prevalence of Motoric Cognitive Risk Syndrome [4].

It is very well-known that aging is associated with a systemic light increase in inflammatory markers, to such an extent that some authors have proposed the term “inflammaging” [96,97]. Over the years it has been shown that the dopaminergic system is particularly prone to be negatively affected by an increase in inflammatory cytokines, also due to the tubero-infundibular system, directly linked to systemic changes in inflammatory cytokines [98,99].

Given the interdependence between the motor and cognitive system and the predominant role of the connection of the reward system, a new pathophysiological mechanism has been hypothesized. Aging causes increased inflammation which acts directly on dopaminergic function, making it dysfunctional. A dysfunction of the dopaminergic system directly compromises the sensorimotor system and the cognitive system but above-all the reward system, making the elderly patient more vulnerable to the development of depression [100,101].

### 3.3. The Loop between the Reward System, Depression, and Mild Cognitive Impairment in Older Persons

Apathy and depression almost coincide in the elderly patient with common pathways and mechanisms. The progressive social isolation of elderly patients and a loss of interest in hedonic activities due to a dysfunction of the reward system are the main determinants of these conditions. In a world that is not ready to deal with the huge wave of patients with dementia-related diseases, very effective preventive strategies are needed. From this point of view, the link between reward system dysfunction, depression and dementia is trivial.

The presence of a neuropsychiatric disorder in an elderly patient facilitates the conversion to dementia. In a cohort study of over 2700 people with MCI (Mild Cognitive Impairment) during a 9-year follow-up period, half the patients who exhibited apathy at the start of the study converted to dementia in about three years, compared to nearly four and a half years for patients without apathy. Furthermore, apathy is a predictor of stronger irritability and depression [102].

In a meta-analysis of 16 studies on over 7000 elderly people, the presence of apathy is a risk factor for conversion to dementia in elderly people with MCI, with a risk of converting to dementia twice as high as in non-apathetic patients [103].

Interestingly, apathy is a risk of conversion to dementia even in cognitively intact patients. In a study of 3500 elderly patients over 70 followed for 7 years, apathy, rather than depression, was a risk factor for conversion to dementia, with a 20% increased risk compared to non-apathetic elderly people [104].

In the same cohort, the effects of apathy on cognitive decline and mortality were dose-dependent, with a double risk of dementia and mortality in those who at the beginning of the study had a more marked apathy [104].

In a study of nearly 1000 elderly people followed for 3 years, apathy determined, in addition to cognitive impairment with a greater progression towards frailty, a slowing gait speed and disability in the activities of daily living [105]. The effect was dose dependent, with a three times higher risk of disability in patients who had a higher degree of apathy at the start of the study. The concordance between neuropsychiatric symptoms and conversion to dementia leads researchers to stratify patients with mild cognitive impairment to try to identify the symptomatic pattern that most predisposes to conversion to dementia. In one study, patients with more marked behavioral disturbances (including apathy) were found to convert to dementia nearly three times more (2.69; 95% CI: 1.12–2.70) than those without neuropsychiatric symptoms [106]. Finally, apathy increases the risk of developing Motoric Cognitive Risk Syndrome by more than two times [107].

Considering the evidence at our disposal, we intend to propose a hypothetical loop that connects the reward system, depression and dementia (Figure 2). Virtually a loss of rewards with aging predisposes to depression and cognitive impairment which in turn predisposes to reactive depression, thus triggering a vicious circle that increases the likelihood of elderly patients with an initial and mild cognitive impairment to convert to dementia. The trigger of this pathological loop is a dysfunction in the dopaminergic system.

This model makes it possible to focus attention on the prevention of dementia, shifting the preventive paradigm from the individual to the community. We need social systems that prevent the isolation of the elderly; this approach has proved very useful in the prevention of dementia.

In a review of prospective studies that evaluated the association between social networks and dementia, apathy and social isolation were important factors of conversion to dementia. Instead, being part of social groups and being involved in multiple enjoyable and productive activities were protective factors [108].

In a study of 4500 elderly people followed for three years, the cognitive decline was inversely correlated to the number of social groups in which the subjects were involved. In the same study, being involved in groups in which specific activities were carried out, such as volunteering, was further protective with respect to the risk of cognitive impairment [109].

In a prospective study of 3000 elderly people followed for almost 20 years it was shown that it is not so much the size of the social network that determines an improvement in cognitive functions, but its complexity: having more children, grandchildren, colleagues, and so forth allows the brain to adapt to different social contexts with a benefit as regards the maintenance of its function [110]. We have already seen how being together with the other is in itself an activating factor of the rewards system and more and more evidence indicates that being socially involved even in old age is protective in preventing disability related to cognitive decline [111,112].

Even if the unidirectionality of the pathophysiological loop proposed by us seems unambiguous, it is nevertheless to be kept in mind that an inverse mechanism could be established where the cognitive decline exacerbates the motor dysfunction. Although this possibility remains a viable hypothesis, most models show that dementia develops over a period of many years, and before it develops, in a prodromal phase there is indeed a deterioration of motor functions. Probably the motor functions begin to be compromised before the cognitive functions, the evidence in the literature to support this thesis is still scarce but in some studies the loss of motor functions is one of the main markers of the future loss of cognitive functions [113]. If the loss of motor functions manifests itself as falls, a precipitating factor towards cognitive decline could also be the development of secondary depression [114,115].

It should also be considered how a deflection of mood can be directly caused by a loss of motor functions, regardless of age. In this context, correct motor rehabilitation prevents depression and if this develops, its treatment is beneficial in the recovery of motor functions [116].

Finally, even if it is difficult to establish a temporality of onset between depression and slow gait speed, some evidence indicates that patients with a slow walking speed are more predisposed to the development of depressive symptoms and even in a trial it was shown that the use of L-DOPA, with its known beneficial effects on motor functions, may relieve depressive symptoms [117,118].

## 4. Conclusions

Considering the evidence at our disposal, we intend to propose a hypothetical loop that connects the reward system, depression and dementia (Figure 2). It can be hypothesized that the age-related impairment predisposes to depression and cognitive impairment and subsequent reactive depression. The cascade of events may increase the likelihood of elderly patients with an initial and mild cognitive impairment to convert to dementia. The trigger of this pathological loop is the dysfunction in the dopaminergic system.

This model shifts the attention to the prevention of dementia, from the individual to the community. We need social systems (mild cognitive impairment friendly communities) that prevent the isolation of the elderly, an approach that was shown to be very useful for the prevention of dementia.

Moreover, there is the need to move attention from single components (motor system, cognitive system, cognitive decline) to the entire individual. This type of approach has been borrowed from the geriatric comprehensive assessment integrating multiple domains.

For instance, Motoric Cognitive Risk Syndrome is frequently associated with polypharmacy, so one of the successful strategies for preventing dementia can necessarily include deprescribing [113]. While waiting for new models for studying cognitive impairment or new therapeutic approaches counteracting the progression of cognitive decline, we believe that an integrated view of the patient is the best way to prevent dementia. Thus, it should be emphasized that although differences in the dopaminergic system implicated in rewards among the different genders have been highlighted in some preclinical models, in humans there is still little evidence to this effect [119]. Further studies are needed to identify the correct categories of patients at risk first and further studies will then serve to identify the best preventive strategies for cognitive decline.

## Figures and Tables

**Figure 1 biomedicines-10-00808-f001:**
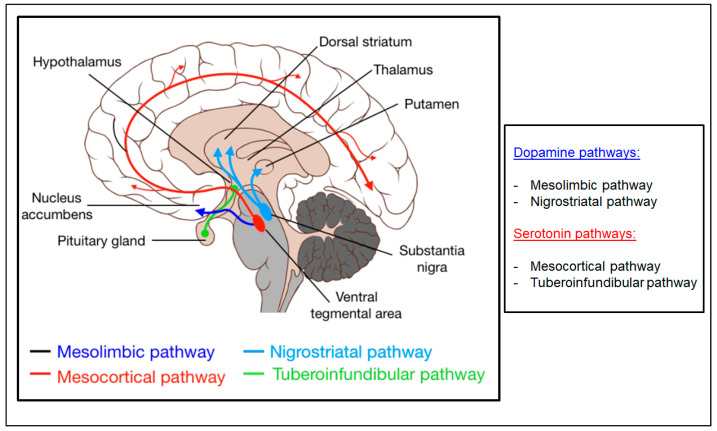
The main dopaminergic pathways in the central nervous system.

**Figure 2 biomedicines-10-00808-f002:**
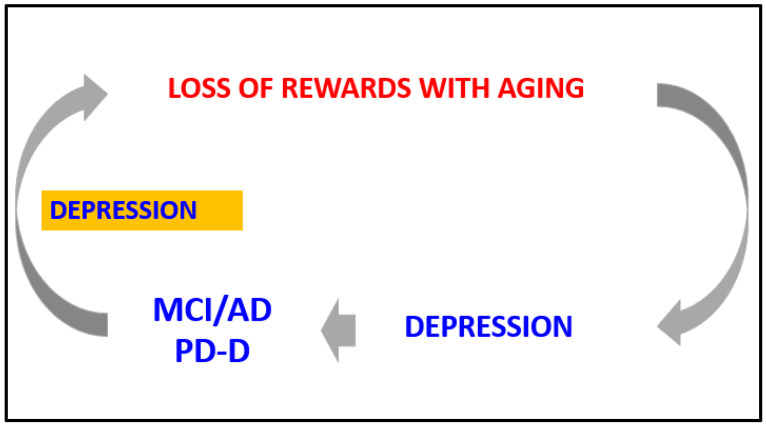
Hypothetical loop explaining the potential link between age-related changes in rewards system, depression, Mild Cognitive Impairment (MCI), Alzheimer’s Disease (AD) and Parkinson’s Disease (PD) Dementia.

## Data Availability

Not applicable.

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
