# Peer review of "Reward System Dysfunction and the Motoric-Cognitive Risk Syndrome in Older Persons"

_biomedicines, 2022, doi:10.3390/biomedicines10040808_

Round 1
Reviewer 1 Report
The manuscript by Lauretani and co-authors is an overview of the reward system in elderly patients. It convincingly argues the potentiality of conceptualizing the cognitive and motor disturbance of these patient from a more integrate point of view, i.e., centered on the reward system. In particular, the existence of a loop between the age-dependent alterations of the latter and cognitive impairment is proposed, hinting at possible underlying mechanisms.
The article is nicely written, and its innovative viewpoint deserves adequate attention for the implications it entails for clinicians. For this reason, I will limit myself to a few observations, mainly aimed at integrating some points of the text.
Throughout the manuscript, the authors have never mentioned the possibility of reverse causation as interpretative bias, yet it must always be borne in mind. For example, multiple falls in the elderly patient can be caused by cognitive decline, although it cannot be excluded that are the consequences of the fall to expose the subject to social withdrawal, implying fewer meaningful contact with others, and reinforcing the vicious circle conducive to dementia.
Another crucial aspect, in my opinion, is the exact timing of events. Do elderly patients develop dementia before or after motor disability? Did depression occurred before or after a slowed walking speed, or do they develop them simultaneously, thus referring to a common denominator/mechanism? Literature is still scarce in this regard. The results of the Einstein Aging Study show, for example, that multiple falls precede the onset of cognitive decline (PMID: 35290430).
Finally, another aspect that could have been addressed is the potential gender differences in the responsiveness of the reward system. While in model organisms a gender difference in the dopamine system implicated in the reward system has been demonstrated (PMID: 29946108), human studies in this field are still rare. An interesting possibility is that the control of the reward-related dopaminergic brain activation depends on age-dependent epigenetic modifications that the new "omics" technologies will make increasingly investigable.
Minor remarks:
Language is very accurate, the only exception being represented by "tegumental" instead of "tegmental", in the abstract (line 15) which must be written as "ventral tegmental area". Also in the abstract, “the existence of a loop” on line 21.
Page 1, line 28. Delete the genitive in "For years"
Page 2, lines 9-10; page 6, line 35. In formal discourse, “more and more” is unacceptable. Please use a synonym more suitable for a scientific text, e.g., “increasing evidence”, “accumulating evidence” or similar. Perhaps this also applies to “food for thought” (page 3, lines 21-22), and “as often happens” (page 4, line 47).
Page 2, line 36: “to the balance”.
Page 3, Figure 1. The labels near the arrows are too small.
Page 4, line 41. This sentence is practically a duplicate of the preceding one.
Page 5, line 39: Perhaps “downregulation”, “reduced activity” instead of “reduction”.
Page 5, lines 52-53: “α-synuclein”.
Page 6, line 24: “enters by right into this context”.
Page 10, line 6: “more than twice”.
Page 10. The first 7 lines of the conclusion (lines 5-11) are a simple paraphrase of the lines on page 10, lines 7-12. If possible, it would be better to suppress or shorten them.
Author Response
Reviewer 1:
The manuscript "REWARD SYSTEM DYSFUNCTION AND THE MOTORIC-COGNITIVE RISK SYNDROME IN OLDER PERSONS" by Lauretani et al. shows a hypothesis that there is a causal relationship of reward system dysfunction, depression, and neurodegenerative diseases in elderly humans. The Authors describe the functioning of the reward system thoroughly in the context of pathologies, such as apathy, depression, or other cognitive impairments. They also show convincing literature data supporting their hypothesis about the existence of the loop linking reward system impairment, depression, and dementia. Such a view can strengthen the support for holistic therapies in preventing neurodegenerative diseases.
There are a few minor issues to be resolved:
1.There are some incomplete sentences in the text, e.g., Abstract, line 13, or on page 4, line 40.
2.Some parts of the manuscript contain one-sentence paragraphs. It makes the reading difficult because the flow of the text is disturbed.
3.The abbreviation "MCI" should be explained in the place of the first use.
4.The brain model in Figure 1 seems not to be drawn by the Authors. The Google search by picture allows finding an analogous model in some websites. Do Authors have copy-rights to use this picture?
5.The reviewer discourages using the word "new" in the context of their hypothesis (page 10: lines 7 and 17, Figure 2, page 11: lines 5 and 12). Although it may be true that this hypothesis/model is new, it is always not possible to be 100% sure about it. Thus, it is better to avoid this word and leave just "hypothesis" or "model."
- Thank you for your comments. We changes the article according to your suggestions. We also changed the Figure 1.
Reviewer 2 Report
The manuscript "REWARD SYSTEM DYSFUNCTION AND THE MOTORIC-COGNITIVE RISK SYNDROME IN OLDER PERSONS" by Lauretani et al. shows a hypothesis that there is a causal relationship of reward system dysfunction, depression, and neurodegenerative diseases in elderly humans. The Authors describe the functioning of the reward system thoroughly in the context of pathologies, such as apathy, depression, or other cognitive impairments. They also show convincing literature data supporting their hypothesis about the existence of the loop linking reward system impairment, depression, and dementia. Such a view can strengthen the support for holistic therapies in preventing neurodegenerative diseases.
There are a few minor issues to be resolved:
- There are some incomplete sentences in the text, e.g., Abstract, line 13, or on page 4, line 40.
- Some parts of the manuscript contain one-sentence paragraphs. It makes the reading difficult because the flow of the text is disturbed.
- The abbreviation "MCI" should be explained in the place of the first use.
- The brain model in Figure 1 seems not to be drawn by the Authors. The Google search by picture allows finding an analogous model in some websites. Do Authors have copy-rights to use this picture?
- The reviewer discourages using the word "new" in the context of their hypothesis (page 10: lines 7 and 17, Figure 2, page 11: lines 5 and 12). Although it may be true that this hypothesis/model is new, it is always not possible to be 100% sure about it. Thus, it is better to avoid this word and leave just "hypothesis" or "model."
Author Response
Reviewer 2:
The manuscript by Lauretani and co-authors is an overview of the reward system in elderly patients. It convincingly argues the potentiality of conceptualizing the cognitive and motor disturbance of these patient from a more integrate point of view, i.e., centered on the reward system. In particular, the existence of a loop between the age-dependent alterations of the latter and cognitive impairment is proposed, hinting at possible underlying mechanisms.
The article is nicely written, and its innovative viewpoint deserves adequate attention for the implications it entails for clinicians. For this reason, I will limit myself to a few observations, mainly aimed at integrating some points of the text.
Throughout the manuscript, the authors have never mentioned the possibility of reverse causation as interpretative bias, yet it must always be borne in mind. For example, multiple falls in the elderly patient can be caused by cognitive decline, although it cannot be excluded that are the consequences of the fall to expose the subject to social withdrawal, implying fewer meaningful contact with others, and reinforcing the vicious circle conducive to dementia.
Another crucial aspect, in my opinion, is the exact timing of events. Do elderly patients develop dementia before or after motor disability? Did depression occurred before or after a slowed walking speed, or do they develop them simultaneously, thus referring to a common denominator/mechanism? Literature is still scarce in this regard. The results of the Einstein Aging Study show, for example, that multiple falls precede the onset of cognitive decline (PMID: 35290430).
Finally, another aspect that could have been addressed is the potential gender differences in the responsiveness of the reward system. While in model organisms a gender difference in the dopamine system implicated in the reward system has been demonstrated (PMID: 29946108), human studies in this field are still rare. An interesting possibility is that the control of the reward-related dopaminergic brain activation depends on age-dependent epigenetic modifications that the new "omics" technologies will make increasingly investigable.
- Thank you for your comments. In the new version of the paper we included the suggested papers and comment as suggested by the reviewer
Minor remarks:
Language is very accurate, the only exception being represented by "tegumental" instead of "tegmental", in the abstract (line 15) which must be written as "ventral tegmental area". Also in the abstract, “the existence of a loop” on line 21.
Page 1, line 28. Delete the genitive in "For years"
Page 2, lines 9-10; page 6, line 35. In formal discourse, “more and more” is unacceptable. Please use a synonym more suitable for a scientific text, e.g., “increasing evidence”, “accumulating evidence” or similar. Perhaps this also applies to “food for thought” (page 3, lines 21-22), and “as often happens” (page 4, line 47).
Page 2, line 36: “to the balance”.
Page 3, Figure 1. The labels near the arrows are too small.
Page 4, line 41. This sentence is practically a duplicate of the preceding one.
Page 5, line 39: Perhaps “downregulation”, “reduced activity” instead of “reduction”.
Page 5, lines 52-53: “α-synuclein”.
Page 6, line 24: “enters by right into this context”.
Page 10, line 6: “more than twice”.
Page 10. The first 7 lines of the conclusion (lines 5-11) are a simple paraphrase of the lines on page 10, lines 7-12. If possible, it would be better to suppress or shorten them.
- Thank you for your comments. We change the article according to your suggestions.